# Carbamazepine Gel Formulation as a Sustained Release Epilepsy Medication for Pediatric Use

**DOI:** 10.3390/pharmaceutics11100488

**Published:** 2019-09-20

**Authors:** Saeid Mezail Mawazi, Sinan Mohammed Abdullah Al-Mahmood, Bappaditya Chatterjee, Hazrina AB. Hadi, Abd Almonem Doolaanea

**Affiliations:** 1Department of Pharmaceutical Technology, Kulliyyah of Pharmacy, International Islamic University Malaysia, Kuantan 25200, Malaysia; saeidmezail@yahoo.com (S.M.M.); bdpharmaju@gmail.com (B.C.); hazrina.hadi@gmail.com (H.A.H.); 2School of Pharmacy, PICOMS International University College, Batu Muda, Batu caves, Kuala Lumpur 68100, Malaysia; 3Pharmacy College, Al-Kitab University, Kirkuk 36010, Iraq; sinan.almawla@gmail.com; 4Department of Pharmaceutics, SPPSPTM, SVKM’s NMIMS (Deemed to be University), Mumbai 400056, India; 5IKOP Sdn Bhd, Kulliyyah of Pharmacy, International Islamic University Malaysia, Kuantan 25200, Malaysia

**Keywords:** gel, sustained release, carbamazepine, epilepsy, pediatric

## Abstract

This study aimed to develop a carbamazepine (CBZ) sustained release formulation suitable for pediatric use with a lower risk of precipitation. The CBZ was first prepared as sustained release microparticles, and then the microparticles were embedded in alginate beads, and finally, the beads were suspended in a gel vehicle. The microparticles were prepared by a solvent evaporation method utilizing ethyl cellulose as a sustained release polymer and were evaluated for particle size, encapsulation efficiency, and release profile. The beads were fabricated by the dropwise addition of sodium alginate in calcium chloride solution and characterized for size, shape, and release properties. The gel was prepared using iota carrageenan as the gelling agent and evaluated for appearance, syneresis, drug content uniformity, rheology, release profile, and stability. The microparticles exhibited a particle size of 135.01 ± 0.61 µm with a monodisperse distribution and an encapsulation efficiency of 83.89 ± 3.98%. The beads were monodispersed with an average size of 1.4 ± 0.05 mm and a sphericity factor of less than 0.05. The gel was prepared using a 1:1 ratio (gel vehicle to beads) and exhibited no syneresis, good homogeneity, and good shear-thinning properties. The release profile from the beads and from the gel was not significantly affected, maintaining similarity to the tablet form. The gel properties were maintained for one month real time stability, but the accelerated stability showed reduced viscosity and pH with time. In conclusion, CBZ in a gel sustained release dosage form combines the advantages of the suspension form in terms of dosing flexibility, and the advantages of the tablet form in regards to the sustained release profile. This dosage form should be further investigated in vivo in animal models before being considered in clinical trials.

## 1. Introduction

Carbamazepine (CBZ) is an epilepsy treatment used by patients of different age groups, including pediatrics and especially children below six years old. However, the sustained release dosage is only available as a solid oral dosage form (tablet or capsule). Children below six years old usually have difficulty swallowing solid oral dosage forms [1,2,3]. The United States Food and Drug Administration (FDA) and European Medical Agency (EMA) have encouraged developers to make sustained release formulations for pediatric use [4,5,6]. CBZ can be given as a suspension for children in 2–4 doses per day, but several side effects have been reported due to precipitation of the suspension [7,8,9]. This highlights the need for a new formulation that can provide a sustained release and avoid precipitation of CBZ. This study aimed to achieve these goals. Making a sustained release formulation suitable for children with swallowing difficulties (below six years old) is challenging. Based on EMA guidelines, currently only solution and suspension formulations are suitable for this age group [4,5,6]. While a sustained release form is difficult to develop, the suspension dosage form was considered.

Carbamazepine (CBZ) is an anticonvulsant drug used for the treatment of neuralgia, trigeminal, epilepsy, and bipolar disorder. CBZ is a white-yellowish, bitter tasting powder that is insoluble in water [10,11]. The FDA approved carbamazepine formulations include chewable tablets (Tegretol^®^), a suspension (Tegretol^®^), sustained release capsules (Carbatrol^®^), and sustained release tablets (Tegretol^®^-XR) [7]. Following a twice a day dosage regimen, the suspension provides higher peak levels and lower trough levels than those obtained from conventional tablets for the same dosage regimen. On the other hand, following a thrice daily dosage regimen, the CBZ suspension affords steady-state plasma levels comparable to CBZ tablets given twice a day when administered at the same total mg daily dose [7]. Following a twice daily dosage regimen, CBZ extended-release tablets afford steady-state plasma levels comparable to conventional CBZ tablets given four times a day, when administered at the same total mg daily dose [7]. CBZ is prescribed for children under 6 years of age at a dose of 10 to 20 mg/kg/day twice a day or three times a day. The dose is increased weekly to achieve an optimal clinical response, and administration may be increased to three times a day or four times a day. As such, the suspension dosage form allows dosing flexibility. However, it lacks the property of sustained release. This is available only in the solid oral dosage forms, like tablets and capsules. There is still a need for a sustained release CBZ formulation to be developed that provides both the required release profile and dosing flexibility.

The International Conference on Harmonisation of Technical Requirements for Registration of Pharmaceuticals for Human Use (ICH) has classified five different age groups: preterm new-borns infants, new-born infants (0–27 days), infants and toddlers (28 days–23 months), children (2–11 years), and adolescents (12 to 16–18 years, depending on the region) [12,13]. Due to the anatomy of their buccal cavity development, this young population is usually unable to swallow capsules or tablets. However, children may be able to swallow small tablets but not larger tablets. According to the Guideline on Development of Medicines for Paediatric Use, the acceptability of the tablets depends on their age and the size of the tablets [5,6]. For children from 2 to 5 years old (pre-school), the preferred dosage forms are solutions and suspensions, with the ability to swallow tablets smaller than 5 mm [5]. However, overdose caused by suspension sedimentation is one of the common reported issues with CBZ suspensions.

Most of the CBZ sustained release formulations have been developed as tablets. Only a few formulations have been attempted for paediatric use. Among them, nanoparticles and microparticles are the most reported, despite the authors not claiming their suitability for the paediatric population [14,15,16,17,18,19,20]. However, as multiparticulate systems with a size that can be swallowed by children younger than six years old, those dosage forms might be useful [21].

There are several types of polymers or coating materials, such as celluloses (like ethyl cellulose, methyl cellulose, carboxymethyl cellulose), gums (like gum Arabic), carrageenans (kappa carrageenan, iota carrageenan, and lambda carrageenan), and alginates (sodium alginate). Ethyl cellulose (EC) is the most stable polymer among the cellulose derivatives. It can resist alkalis in both concentrated and diluted solutions. It only adsorbs a very small amount of water from the moist air or from aqueous solutions during immersion [22]. There are three types of carrageenan—kappa carrageenan, iota carrageenan, and lambda carrageenan. Iota carrageenan is the only type that shows no syneresis [23]. Sodium alginate is a natural polysaccharide polymer that is soluble in cold and hot water. It is biocompatible, non-toxic, and widely used in pharmaceutical beads, microparticles, nano-particles, and sustained release preparations [24]. Beads of alginate are easily prepared by crosslinking with divalent cations, such as calcium.

Oral gel is a fundamental solution for dysphagic patients and is frequently prescribed for geriatrics [25,26]. The gel has the advantage of preventing precipitation due high viscosity. In addition, when the gel is shear-thinned, it allows flexible dosing because it flows easily upon application of suitable stress, like withdrawal by syringe. Gel formulations have been developed for some drugs already, such as paracetamol [27]. Gel formulations can be designed for immediate release or sustained release [28]. One study described a CBZ oral gel and performed a comparison study between different gelling agents and a stability study for those gelling agents, but the release was almost completed in 1 h (immediate release dosage form) [25,26]. There is no existing report on a sustained release gel formulation for CBZ.

In this study, a CBZ sustained release formulation suitable for pediatric use, especially for children below six years old, was developed based on multiparticulate systems (microparticles and beads). These particles were embedded in a gel as the final dosage form. The in vitro release properties were compared with those of commercial CBZ sustained release tablets.

## 2. Materials and Methods

### 2.1. Materials

Carbamazepine (CBZ) was purchased from Anuja Healthcare Limited (Punjab, India). Ethyl cellulose (EC) was bought from DOW chemicals (Louisiana, Greensburg, PA, USA). Polyvinyl alcohol (PVA), dichloromethane, and other chemicals were procured from Merck (Hohenbrunn, Germany). Sodium alginate was purchased from FMC (Philadelphia, PA, USA). Calcium chloride was obtained from Merck (Darmstadt, Germany). Iota carrageenan and honey powder were sourced from the Modernist Pantry (Portsmouth, NH, US). Propyl paraben was bought from Parchem (New Rochelle, NY, USA).

### 2.2. Preparation and Characterization of CBZ-Loaded Microparticles

CBZ was encapsulated in ethyl cellulose microspheres using the solvent evaporation technique. Briefly, 200 mg of CBZ and 200 mg of EC were dissolved in 10 mL dichloromethane to create the oil phase. The aqueous phase was prepared by dissolving PVA in water at 1% *w*/*v* concentration. Ten mL of the oil phase was added to 30 mL of the aqueous phase (ratio of 1:3) and mixed at 6000 rpm using a homogenizer for 180 s. The mixture was added to 120 mL of the dispersion medium of distilled water (ratio 1:3 emulsion to dispersion medium) and stirred for 3 h at room temperature to evaporate the organic solvent. Microparticles were collected by filtration then washed three times with distilled water [29,30].

The particle size of the CBZ-loaded microparticles was evaluated by suspending the microparticles in distilled water, followed by the measurement using a laser diffraction particle sizer (BT-9300H, Liaoning, China). Polydispersity of the microparticles was evaluated by calculating the span value (Equation (1)).
Span value = (D90 − D10)/D50,(1)
where D10: 10% of the particles are smaller than this diameter; D50: 50% of the particles are smaller than this diameter; D90: 90% of the particles are smaller than this diameter [31].

The particle morphology was observed under a light stereomicroscope (Nikon SMZ745, Tokyo, Japan) attached to a Nikon special lens (Nikon DS-Fi2, Tokyo, Japan). 

The presence of CBZ in the microparticles was confirmed by attenuated total reflectance Fourier transform infrared (ATR-FTIR) spectroscopy. Ethyl cellulose, CBZ, and CBZ microparticles were scanned in the range 4000–400 cm^−1^ using a Perkin-Elmer FTIR spectrometer (Perkin Elmer Corp., Norwalk, CT, USA).

The amount of CBZ encapsulated in the microparticles was determined by adding 10 mg of microparticles into 100 mL phosphate buffer at pH 6.8 and stirring at 200 rpm using an incubator shaker (INNOVA 4000, GMI, Ramsey, MN, USA) at 37 ± 0.05 °C for 4 h. Then, the sample was filtered (Whatman filter paper grade 1) and analyzed (*n* = 3) at 286 nm using a UV-spectrophotometer (SHIMADZU UV-1800, Kyoto, Japan). Encapsulation efficiency was calculated by dividing the actual amount of CBZ in the microparticles by the theoretical amount.

### 2.3. Encapsulation of CBZ-Loaded Microparticles in Alginate Beads

A quantity of the microparticles equivalent to 200 mg of CBZ was suspended in 5 mL of 2% *w*/*v* sodium alginate solution using a magnetic stirrer. The electrospray apparatus was assembled using a high voltage power supply from Genvolt (Shropshire, UK) and a syringe pump from Shenchen Pump (Baoding, China), and a stainless-steel needle. The microparticle-in-alginate suspension was extruded into 1% *w*/*v* calcium chloride solution using the electrospray at a 1 mL/min flow rate, with a 10 cm distance between the needle head and calcium chloride solution, and 4000 kV. Alginate beads were formed instantly by means of crosslinking between the alginate and calcium ions. The beads were collected using a metal mesh and washed using distilled water.

The particle size of 10 beads was determined using a digital microscope, U500 Shenzhen (Guangdong, China). The sphericity factor (SF) was used as an indication of the shape of the beads and calculated using Equation (2).
Sphericity factor (SF) = (D_max_ − D_min_)/(D_max_ + D_min_),(2)
where D_max_ is the longest diameter and D_min_ is the shortest diameter of the same bead.

The EE of the CBZ in the alginate beads was measured to ensure the loading of CBZ-loaded microparticles inside the beads. An indirect method was utilized by measuring the un-encapsulated amount of CBZ. Un-encapsulated CBZ leaks into calcium chloride solution either as soluble CBZ or as microparticles. Soluble CBZ was quantified using a spectrophotometer at a 286 nm wavelength. Meanwhile, measurement at 600 nm was also conducted and used to measure the turbidity in the calcium chloride solution resulting from the presence of microparticles that have not been embedded in the beads.

### 2.4. Loading of CBZ Alginate Beads in a Gel Vehicle

The gel vehicle was prepared by heating 40 mL of distilled water up to 90 °C, followed by addition of the gelling agent under continuous stirring. After complete dissolving and cooling, propyl paraben sodium was added as preservative then the solution was made up to 50 mL with distilled water. Two types of carrageenan (iota or kappa) at two concentrations (1 and 2.5% *w*/*v*) were used as gelling agents. CBZ alginate beads containing 2000 mg CBZ (about 50 mL as volume) were added into the gel vehicle at 40 °C, then left to cool down at room temperature to produce the gel formulation. The final gel formulation was prepared in two ratios—1 to 1 and 1 to 2 volume ratios of beads to iota carrageenan jelly. Therefore, the volume fraction of the alginate beads in the final gel was 1/2 (for the 1 to 1 ratio) and 1/3 (for the 1 to 2 ratio).

### 2.5. Characterization of CBZ Oral Gel

#### 2.5.1. Physical Appearance and Syneresis

Visual observation of the fabricated CBZ-oral gels was undertaken to assess their clarity, smell, texture, and the presence of any foreign particles. The texture of the prepared gels was tested by rubbing them between two fingers to check their grittiness and stickiness.

Water separation from the gel is known as syneresis, which is a common problem associated with jellies during their storage [32,33,34] (Brinker and Scherer, 2013). Iota and kappa carrageenan gels at two different concentrations (1% and 2.5% *w*/*v*) were stored at room temperature (25 ± 2 °C) and in the refrigerator (8 ± 3 °C) for 24 h to observe syneresis, if any [25].

#### 2.5.2. Homogeneity

A homogeneity test was carried out to ensure the distribution uniformity of CBZ beads in the gel. Beads prepared from 5 mL alginate suspension were added into 5 mL or 10 mL 2.5% *w*/*v* iota carrageenan gel to yield a ratio of 1:1 or 1:2. Three 1 mL aliquots from the gel (upper, middle and lower regions) were individually dissolved in 100 mL of phosphate buffer (pH 6.8) under continuous shaking at 200 rpm using an incubator shaker (Innova 4000, GMI, Ramsey, MN, USA) at 37 for 4 h. The samples were filtered and analyzed at 286 nm using a spectrophotometer (Shimadzu UV-1800, Kyoto, Japan) [35].

#### 2.5.3. Rheology

Iota carrageenan gelling agent was first tested for rheology using a HAAKE Mars rheometer (Thermo-Scientific, Waltham, MA, USA). The data were then digitally analyzed by Haake Rheo-Win 3.61.0000 software (Thermo-Scientific, Waltham, MA, USA). The test was conducted at 25 °C ± 0.05 °C using a PP35 Ti spindle of 35 mm diameter and 1 mm gap. The test was conducted at an increasing shear rate (*γ*) from 0.01 to 100 s^−1^ at 1 Hz frequency, then a constant shear rate of 100 s^−1^ for 30 s, then a decreasing shear rate from 100 to 0.015 in 120 s [36]. The results were represented graphically as apparent viscosity (*η*)*,* and shear stress (*τ)* vs. shear rate (*γ*). The same test was repeated after loading the CBZ alginate beads in the gel vehicle. Rheological modelling for each of the gels was fitted to either the Herschel Bulkley or Ostwald De Waele model.

### 2.6. In Vitro Drug Release

The sustained release properties of the CBZ gel formulation were evaluated and compared with the commercially available 200 mg Tegretol^®^-XR sustained release tablet (Novartis), by calculating the similarity factor (F2) using Equation (3). In addition, the release profiles of CBZ powder, CBZ microparticles, and CBZ-alginate beads were also compared in order to observe the effect of each preparation step on the release properties.
(3)$$f_{2}=50log\left\{{\left[1+\frac{1}{\rho }\sum_{j=1}^{\rho }{\left(u_{Tj}-u_{Rj}\right)}^{2}\right]}^{-\frac{1}{2}}\times 100\right\},$$
where F2 is the similarity factor, R_t_ and T_t_ are the dissolved cumulative percentage of test and reference samples at selected (t) time points respectively, and n is the number of time points [37,38,39].

The release profile tests were conducted with a USP-II dissolution apparatus using a 900 mL volume of dissolution medium at 37 ± 0.5 °C and a rotating speed of 100 rpm. At a predetermined time, point (1, 3, 6, 12, and 24 h), 5 mL of dissolution medium was taken (*n* = 3) and replaced with an equal volume of fresh medium. The samples were filtered and analyzed spectrophotometrically at the wavelength of 286 nm.

For CBZ powder, CBZ microparticles and CBZ-alginate beads, the dissolution test was adopted from United States Pharmacopeia (USP), since these formulations can be considered solid dosage forms [40]. However, CBZ gel is a new dosage form not described in USP. Therefore, to compare it with commercial formulations, the dissolution was conducted using the two stage general dissolution method (HCl and phosphate buffer stages). 

The Tegretol^®^-XR sustained release tablets containing 200 mg CBZ were immersed in HCl for the first 2 h, then the disintegrated tablet was filtered using Whatman filter paper no. 1. The tablet particles were collected easily, then transferred directly into phosphate buffer (pH 6.8) to continue the dissolution test for up to 24 h. For the CBZ gel, an amount containing 200 mg CBZ was immersed in HCl for 2 h. In another dissolution vessel, a similar quantity was added into the phosphate buffer pH 6.8 and the test was run for 24 h. The readings from 2 h were deducted from all subsequent readings.

### 2.7. Stability Study

The CBZ gel formulation, packed in glass containers, was evaluated for physical stability at 30 ± 2 °C with 70 ± 5% RH (real time stability conditions) and 40 ± 2 °C with 70 ± 5% RH (accelerated stability conditions) for 30 days (Countries, 2005). At different time points (0, 7, 15, and 30 days), various quality attributes were evaluated, including pH, physical appearance, syneresis, viscosity, and assay. The viscosity was evaluated using a Brookfield^®^ viscometer (DVII+, Massachusetts, USA) with a spindle number (CPE51) at room temperature (25 °C ± 5 °C) and a rotating speed of 50 rpm [32].

### 2.8. Statistical Analysis

Statistical analysis was completed using a fractional factorial design for the preparation of all formulations, and a t-test for the stability study of the gel utilizing Minitab software (version 17.1.0; Minitab Inc., State College, PA, USA). When the *p*-value is less than 0.05 the null hypothesis will be rejected, and it will be considered as significant. When *p*-value is more than 0.05, the results will be considered as not significant and the null hypothesis will be accepted.

## 3. Results

### 3.1. CBZ-Loaded Microparticles

CBZ was successfully encapsulated in ethyl cellulose microparticles using the solvent evaporation method. The microparticles exhibited a particle size of 135.01 ± 0.61 µm with a monodisperse distribution (Figure 1), a span value of 1.28, and an encapsulation efficiency (EE) of 83.89 ± 3.98%. ATR-FTIR spectra confirmed the presence of CBZ in the microparticles, where CBZ peaks were clearly seen in the microparticle spectrum as presented in Figure 1c.

### 3.2. CBZ-Loaded Microparticles in Alginate Beads

Alginate beads were formed instantly upon contact of alginate solution containing CBZ-loaded microparticles with calcium chloride solution. The beads were spherical (Figure 2) with a size of 1.4 ± 0.05 mm and a sphericity factor of less than 0.05, which is considered spherical [41]. Spectrophotometric measurements revealed that only 0.38 ± 0.01% of the CBZ was un-encapsulated and dissolved in the calcium chloride solution. No CBZ-loaded microparticles were detected in the calcium chloride solution, whereby the turbidity of the solution did not change. This reveals that the encapsulation efficiency of CBZ in the beads was 99.62 ± 0.01%.

### 3.3. CBZ Oral Gel

#### 3.3.1. Physical Appearance and Syneresis

CBZ gel prepared from iota carrageenan were transparent, non-sticky, of acceptable consistency, and without any feeling of grittiness. No syneresis was seen with any of the iota carrageenan gels, while all the kappa carrageenan gels exhibited syneresis except for the 2.5% gel stored in the refrigerator.

#### 3.3.2. Homogeneity

Table 1 shows the assay results of CBZ from samples taken from different locations of the gel. The average value of the three locations was 100%. The ratio of 1:1 shows very close agreement between the three locations, while in the 1:2 ratio the CBZ was not homogenously distributed. The lower layer had a high concentration of CBZ compared to the upper layer due to precipitation of the CBZ-alginate beads in the bottom of the gel container.

#### 3.3.3. Rheology

The rheology profile for 2.5% *w*/*v* iota carrageenan gel exhibited a non-Newtonian profile, where the shear stress increased non-linearly with the shear rate. The gel fits the Herschel Bulkley model (R = 0.9981) with clear yield shear stress (Figure 3). However, after incorporating CBZ-alginate beads at a 1:1 loading ratio, the gel did not fit any model and exhibited a degree of thixotropy. The rheology profile of the gel containing the beads was disrupted with spikes. This was because the bead size was more than 1 mm and the gap between the upper and lower spindles was 1 mm. When the spindle rotated, it smashed and compressed the beads. Therefore, the measurement of gel rheology was interrupted with the presence of beads in the gel. The overall trend represents the gel where it appeared to have a shear thinning property with a degree of thixotropy, but the spikes in the graph came from the beads.

### 3.4. In Vitro Drug Release

CBZ oral gel exhibited a sustained release profile for 24 h (Figure 4). The release was comparable to the release from commercial Tegretol^®^-XR tablets, with a similarity factor of *f*_2_ = 74. At t50 % the percentage of CBZ release from the microparticles was 71.05%, was 74.12% from the beads, was 68.94% from the gel, and was 68.25% from the commercial Tegretol^®^-XR tablets. These results revealed that the prepared CBZ gel was comparable to the commercial tablet.

Release kinetic fitting is shown in Table 2. CBZ microparticles fitted best into the Higuchi model, suggesting a diffusion controlled mechanism. This was expected as the polymer used in the microparticle preparation was ethyl cellulose, which is not a fast degrading polymer. Therefore, the CBZ will release by diffusing through the microparticle matrix. On the other hand, CBZ beads and CBZ gel fitted best into first order kinetics. The presence of an alginate matrix around the microparticles, then the gel around the beads, slowed down the release.

### 3.5. Stability Study

The appearance and texture of the gel in terms of stickiness did not change during the stability study (Table 3). No syneresis was observed within the stability period. The pH slightly dropped with time in both real time and accelerated conditions, but the change was not significantly different, with a *p*-value of 0.25. Viscosity of the gel stored at 40 °C decreased significantly, while no significant change was observed for the sample stored at 30 °C.

## 4. Discussion

CBZ was first encapsulated in microparticles using an ethyl cellulose polymer, a well-known polymer for sustained release formulations [42,43,44,45,46]. The microparticles exhibited good sustained release properties, comparable to those of commercial tablets that are given twice daily. However, CBZ-loaded microparticles precipitate when suspended in a liquid vehicle. In addition, microparticles, as relatively large particles, give an unpleasant gritty feeling.

To overcome the precipitation issue of the microparticles, gel was proposed as a vehicle for the microparticles. Gels have high viscosity and do not flow in the normal conditions of temperature and stress, so precipitation is avoided. However, the grittiness of the microparticles may not be overcome. In addition, CBZ may slowly leak out from the microparticles into the gel during long-term storage, resulting in loss of the sustained release properties. To solve this issue, another layer of coating was considered, using alginate beads. Upon crosslinking of alginate with calcium ions, rubbery gel beads are formed with no gritty texture. Alginate beads were prepared to encapsulate the CBZ-loaded microparticles. The beads were spherical in shape, within the recommended size for pediatric use (1–6 years old), and effectively entrapped the microparticles inside them. Since the targeted sustained release profile was obtained from the microparticles, the beads should have little effect on the release profile. The required or the target release profile was found to be similar to the release profile of commercial sustained release tablets, as stated in the United States Pharmacopeia (USP39)—10–35% in 3 h, 35–65% in 6 h, 65–90% in 12 h, and not less than (NLT) 75% in 24 h [47]. A kinetic release study showed that the all prepared formulations followed first order release mechanisms, as the coefficient correlation (R^2^) was equal to 0.989 for the CBZ microparticles, 0.976 for the CBZ beads, 0.948 for the CBZ gel, and 0.948 for the Tegretol^®^-XR tablets. The first order model explained that the release of CBZ based on its dosage form was CBZ-concentration dependent. This means that the release of the drug can be increased when the concentration of that drug is increased in the dosage form over time [48,49,50]. The release study confirmed that the release of CBZ from the beads was comparable to that of the microparticles and was still similar to that of the commercial tablet. 

The last step was to suspend the beads in the gel vehicle. This was performed at a temperature slightly higher than the solidifying temperature of the gelling agent. In comparing kappa and iota carrageenan at two concentrations, iota carrageenan gave the needed properties: no syneresis, homogenous, and with shear-thinning properties. During gel preparation the beads tended to precipitate due to the low viscosity of the hot gel. This is why homogeneity was better at a higher concentration of the beads, where the gel amount was just enough to suspend the beads. The shear-thinning property is needed for this medicated gel in order to allow easy dispensing. Iota carrageenan gel fits the Herschel-Bulkley model, which means it has yield stress—the state of the gel does not change until a specific stress is applied. The gel has high viscosity during storage to prevent bead precipitation, but the viscosity decreases upon applying stress like syringe withdrawal. This allows flexible dosing based on body weight, like a normal CBZ suspension. Similar to the beads, the iota carrageenan gel vehicle was found to have no significant effect on the release properties of CBZ-loaded microparticles. Consequently, the sustained release profile from the final formulation (CBZ-loaded microparticles-in-beads-in-gel) was similar to that of the FDA-approved Tegretol^®^-XR tablet. 

The physical stability study showed that the gel formulation maintained its properties for 30 days when stored at 30 °C. However, the gel viscosity decreased when it was stored at 40 °C. This highlights the importance of storage conditions for this formulation. The pH of the gel was slightly acidic after preparation and showed a decreasing trend during the stability period. Using an ANOVA as the statistical analysis, the change in pH value was found to be not significant, possibly due to the large standard deviation of some points. The change in pH might be attributed to the fact that carrageenans are susceptible to acid-catalyzed hydrolysis [51]. Since the gel was slightly acidic after preparation, the hydrolysis of iota carrageenan might occur and accelerate at a higher temperature, leading to a further drop in pH. This explains the faster drop in pH under the accelerated stability condition (40 °C) compared to the real-time condition (30 °C). Such a change in pH is important to avoid. Iota carrageenan undergoes rapid and extensive loss of viscosity and gelation potential when solutions below pH 5 are heated. The loss of viscosity and gelation potential is primarily due to cleavage of the (1→3) glycosidic linkages [51]. For the gel dosage form, viscosity is critical and should be maintained. The pH of the formulation can be controlled by adding suitable buffering systems.

Based on these preliminary results of the stability study, the gel might need to be stored in the fridge (5 ± 3 °C). A scale up study is recommended to ensure suitability of the preparation method for large scale production. Further investigations are needed, but they are recommended to be done for a larger scale preparation.

## 5. Conclusions

A carbamazepine sustained release new dosage form designed for pediatric use was prepared and analyzed in this study. This dosage form was based on encapsulation of carbamazepine in microparticles to obtain the required sustained release profile. The microparticles were then embedded in alginate beads, which, in turn, were suspended in iota carrageenan gel. The developed formulation has the advantages of a suspension formulation, that is, flexible dosing and being easy to swallow. It also overcomes the issue of carbamazepine precipitation that is seen in the suspension formulation, which leads to overdose. Carbamazepine sustained release gel has the potential to make the advantages of a sustained release dosage form to pediatric patients accessible, especially children below six years old who have no current option for such a formulation.

## 6. Patents

The gel as dosage form was patented under the file number (PI 2017704458/Malaysia), under the patent name “A sustained-release drug composition”.

## Figures and Tables

**Figure 1 pharmaceutics-11-00488-f001:**
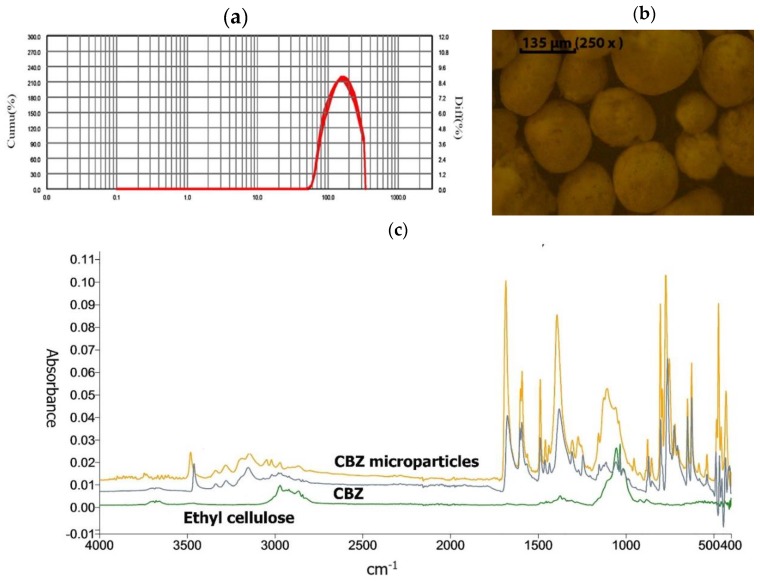
CBZ-loaded microparticle characterization. (**a**) Particle size distribution measured by laser diffraction, (**b**) morphology of the particles under a light microscope, (**c**) ATR-FTIR spectra of ethyl cellulose, CBZ and CBZ-loaded microparticles.

**Figure 2 pharmaceutics-11-00488-f002:**
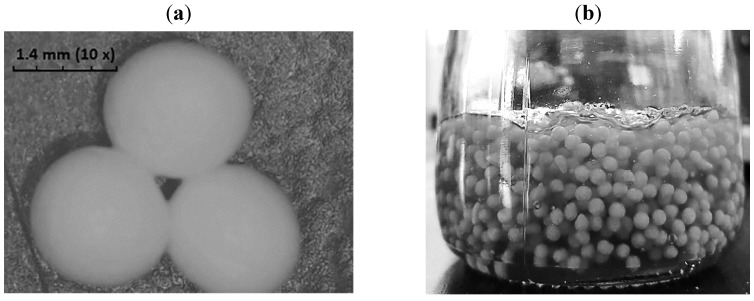
(**a**) CBZ-alginate beads, (**b**) CBZ-gel prepared at a 1:1 ratio of beads to iota carrageenan gel.

**Figure 3 pharmaceutics-11-00488-f003:**
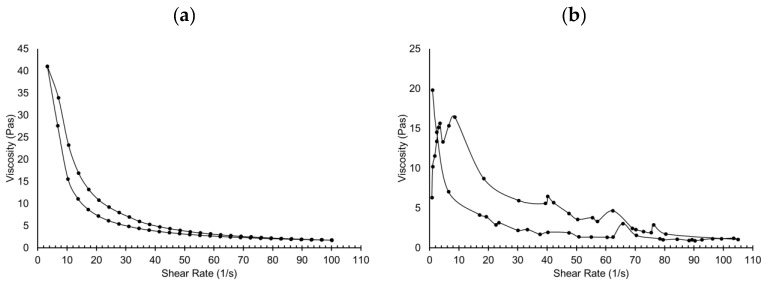
Rheology profile of 2.5% *w*/*v* iota-carrageenan gel, (**a**) without CBZ-alginate beads, (**b**) with CBZ-alginate beads.

**Figure 4 pharmaceutics-11-00488-f004:**
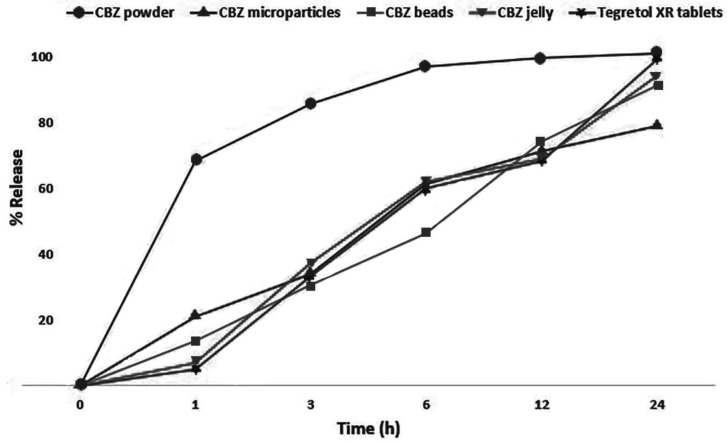
In vitro CBZ release (mean ± 1.86, *n* = 3) from CBZ gel in comparison with Tegretol^®^-XR tablets, CBZ powder, CBZ microparticles, and CBZ beads.

**Table 1 pharmaceutics-11-00488-t001:** Homogeneity of CBZ gel.

Sample Location	Beads to Gel Ratio	CBZ Assay (%)
Up	(1:1)	100.9 ± 1.3
Middle	(1:1)	99.9 ± 1.1
Bottom	(1:1)	99.2 ± 1.9
Up	(1:2)	65.4 ± 2.4
Middle	(1:2)	96.5 ± 3.3
Bottom	(1:2)	138.0 ± 4.5

**Table 2 pharmaceutics-11-00488-t002:** Fitting of the release profiles into different kinetic models.

Model	R2			
	CBZ Microparticles	CBZ Beads	CBZ Gel	CBZ Tablets (Tegretol^®^-XR)
Zero Order	0.7190	0.8828	0.8143	0.8601
First Order	0.8555	0.9968	0.9759	0.9301
Higuchi	0.9222	0.9842	0.9482	0.9591
Korsmeyer Peppas	0.6973	0.7677	0.8121	0.8457
Hixson Crowell	0.8116	0.9760	0.9527	0.9748

**Table 3 pharmaceutics-11-00488-t003:** Stability study of CBZ gel.

Test	Initial (0) Days	7 Days	15 Days	30 Days
**Real Time Conditions (30 ± 2 °C/70 ± 5% RH)**				
Appearance	Transparent with embedded white beads	No change	No change	No change
Syneresis	No	No	No	No
pH	5.7 ± 0.0	5.4 ± 0.1	5.1 ± 0.0	4.5 ± 0.5
Viscosity (mPa.s)	644 ± 271	552 ± 239	676 ± 173	899 ± 208
**Accelerated Conditions (40 ± 2 °C/70 ± 5% RH)**				
Appearance	Transparent with embedded white beads	No change	No change	No change
Syneresis	No	No	No	No
pH	5.8 ± 0.1	5.4 ± 0.0	4.6 ± 0.1	5.0 ± 1.8
Viscosity (mPa.s)	357 ± 37	325 ± 5	253 ± 92	180 ± 22

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
