# Peer review of "Carbamazepine Gel Formulation as a Sustained Release Epilepsy Medication for Pediatric Use"

_pharmaceutics, 2019, doi:10.3390/pharmaceutics11100488_

Round 1

Reviewer 1 Report

The authors describe the preparation and characterization a sustained release formulation of carbamazepine. The idea of the complex system is interesting but the manuscript has serious shortcomings. The described syntheses and experiments are not reproducible and discussion of the results is sloppy. The English of the manuscript must be improved.

Some comments:

– What is the novelty in your manuscript compared to ref. 16 (Formulation development and evaluation of novel oral jellies of carbamazepine using pectin, guar gum, and gellan gum)?
– Methodes “should be described with sufficient detail to allow others to replicate and build on published results. New methods and protocols should be described in detail while well-established methods can be briefly described and appropriately cited.”
> None of the steps can be reproduced, because data is missing. E.g. in lines 91-92: the volume of PVA solution is missing; in line 94: the temperature of evaporation; lines 107-108: volume of sodium alginate solution.
> Equation 2 is incorrect, see your own ref. 29.
> Description of rheological analysis is sloppy. There are too many (incorrect) test descriptions compared to the number of results.
> I would suggest to provide more detailed information on the drug release study. Exact mass and drug content of formulations. How was the "jelly formulation” and the Tegretol tablet placed in the vessel? Did you do any measurement in the 0.1 M HCl? What is the frequency of use in the case of used Tegretol tablet? The release curves are similar in the case of two formulas but is it support the once daily administration?
> I recommend to merge the methods “Physical appearance and syneresis” and “Stability Study”. Discuss the results together.
- As the authors wrote: carbamazepine is insoluble in water. What is the solubility in phosphate buffer (pH = 6.8) and in HCl solution (pH = 1.2). Did you calculate with the effect of the polymer matrix during the absorbance measurement?
– Please use “gel” instead of “jelly”. Sometimes viscous liquid should be the correct term in the manuscript.
– Quality of the pictures is below the level of any scientific paper. I recommend reading the following website:
http://abacus.bates.edu/~ganderso/biology/resources/writing/HTWtablefigs.html
> Fig. 1a: What is the role of this diagram?
> Fig. 2b: Four pictures are unnecessary. Where are the scale bars?
> Figure legend of Fig. 2.: Which kind of carrageenan was used?
> Table 1: term of “Beads: jelly ratio” is incorrect.
> Figure 3. Unusable diagrams. E.g. the points in the diagrams should not be connected by a line in a dot-to-dot fashion. Do not connect the dots when the measurements were made independently. Calculate the best-fit curve!
> I do not understand the scale of x axis. The diagram is misleading.
> Fig. 5: These diagrams are not necessary. Why did you ignore data points after 6 hours? Why did you use only one data series? Did you try to fit a line according to zero order kinetic?
– I think pH change was significant during the stability test (Table 2). Statistical analysis should not override reality. How can you explain the pH change (lines 297-299). Why is it a problem?
– Please, reconsider the use of the number of decimals in the case of encapsulation efficiency, “assay” (Table 1), viscosity (Table 2), etc.
– The first two paragraphs of the discussion is a kind of introduction. I think this repeat is unnecessary and basically this part is written better than the introduction.
– Please explain the “needed sustained release profile”.
– Did you determine of any parameters of Herschel–Bulkley model?
– What is the volume fraction of the alginate beads in the carrageenan gel? Please support in more detail your statement about physical stability.
– Please have your manuscript checked by a native English speaker. I think it is essential for the publication. Several parts of the manuscript simply impossible to understand because of bad English (e.g. lines 46-54) or in line 245: “CBZ as an epilepsy treatment…”. Correct spelling errors before submission (e.g. “multiarticulate systems”, equation 2, “obd=served”, etc.)
– References: some of the citation remained in the text from a previous submission (e.g. Liu et al., 1997).

Reviewer 2 Report

In this manuscript, a new carbamazepine (CBZ) jelly was constructed. The CBZ jelly is mainly using ethyl cellulose microspheres as polymer carrier, forming bead structure in the presence of sodium alginate, and finally using iota carrageenan as gelling agent to form CBZ jelly, which is an interesting finding.

However, there are several concerns.

1)      Line 31 on page 2, it states that "There is no report on jelly formulation for CBZ." But there is a report on CBZ jolly form, please check it. (Prakash K. Formulation development and evaluation of novel oral jellies of carbamazepine using pectin, guar gum, and gellan gum[J]. Asian Journal of Pharmaceutics (AJP): Free full text articles from Asian J Pharm, 2014, 8(4): 241-249. )

2)      The introduction paragraph should be supplemented with an introduction of the related raw vetors commonly used to prepare jelly formulation, and explain on why ethyl cellulose microspheres, sodium alginate and iota carrageenan were selected as CBZ jelly formulation vectors.

3)      It might be suggested to delete the yellow line on the right side and the lower side of Figure 1. The description of the particle size diagram of Figure 1a does not indicate which particle size diagram it is. A text description of the abscissa is missing. And it is recommended to change the abscissa value in Figure 1a to be in Log format, or the drawing should not be stretched.

4)      Figure 1b. Why the scale bar is same with different magnifications (250 x, 300 x, 400 x)?

5)      The research process and schematic diagram on the optimal ratio of vectors can not be found. The data of drug loading, pH, viscosity and dispersion under different ratios of vectors should be compared, and the optimal ratio of vectors should be selected.

6)      The representation figure that can prove the load of CBZ on ethyl cellulose microspheres is missing.

7)      It is recommended to include the particle status of CBZ-loaded microparticles and CBZ-loaded microparticles in alginate beads detected with digital microscope.

8)      The result of CBZ in the beads detected by Spectrophotometric measurements and the calculation formula of encapsulation efficiency of CBZ in the beads were missing.

9)      "CBZ jellies prepared from iota and kappa carrageenans were transparent (Fig. 2)" on line 3 of page 6 is not reflected in figure 2. Please verify whether the picture is correct.

10)   The picture in figure 3 is un-normally stretched, and the box on the left-upper part of Figure 3a is imcomplete.

11)   The representation of small graphs in the figure should be unified with "A, B, C." Or "a, b, c.".

12)   The analysis details of each figure should be clarified.

13)   It could be seen from figures 4 and 5 that the release rate of the prepared CBZ jelly is not much different from that of the commercial tablet, which did not reflect the advantage of sustained release of CBZ jelly.

14)   On page 9, line 6, "obd=served" should be changed to "observed"

15)   Because CBZ jelly is used in children, it might be worthy to do a more stricter test?

Reviewer 3 Report

Dear authors,

Overall, the manuscript is well written and interesting for the reader of Pharmaceutics. There are just some minor comment that should be addressed before publication:

-Some English spelling errors should be corrected such as: through instead of trough (line 44), lack of a verb in line 59... obd=served (line 242)...

-Regarding the materials and methods, dichloromethane has been used as solvent which is quite toxic. Have you tried to prepared the microparticles using a more friendly solvent? Have you quantified the ppm of DCM left over in the particles?

Why do you filter the samples using a Whatman filter paper grade 1? I dont think is a suitable filter to separate the encapsulated from the unencapsualted drug within the microparticles. A 0.45 um filter should be used instead.

Is the sphericity equation correct? It is not clear in line 192 when authors refers to sphericity factor to be 0.05 and they consider that the microparticles are spherical. Should not be the opposite? The higher the sphericity factor indicates a more spherical particles?

Why two different rheometers were used for the rheology and the stability studies?

-Regarding the result section: a PDI (polydispersity index) or Span value should be calculated for the microparticles.

-Why do you think that the rheology profile of figure 3 for the loaded alginate beads is so ackward? Was the measurement performed correctly and in triplicate?The results do not make sense, how can you explain this profile?

Figure 5 is not needed to be included in the manuscript, a small table comparing different type of mechanistic release profile would be more suitable. Also other release mechanisms should be investigates, as authors focused in first order, but zero order may be a better model to explain the release of their drug.

-it is worthy to investigate in more detail why the pH and viscosity after 30 days of storage suffer a significant drop in the values. Viscosity needs units in Table II.

-Statistical analysis should be performed in all the results showed by the authors.

-Scale-up of the formulation should be commented in the discussion.

Round 2

Reviewer 1 Report

I accept the improvements of the Introduction.
I accept the improvements in the paragraph of “Materials and Methods”. But the equation of similarity factor is still incorrect. Use the formula (eq. 4) of Ocana’s paper (ref. 39. in your revised manuscript). Add the calculation formula of encapsulation efficiency to the manuscript, too.

What is the frequency of use in the case of used Tegretol tablet (once or twice a day)? The release curves are similar in the case of two formulas (Tegretol and the developed formula) but is it support the once daily administration as it written in the title of the manuscript?

I do not suggest to use the term “jelly”, I prefer the use of “gel”.

Redraw Figure 3 by yourself if the built-in software is not appropriate. Viscosity vs. shear rate curves are enough and better to show shear thinning and the degree of thixotropy. Eliminate the negative viscosity. Your diagrams should be more uniform in appearance! By the way, measuring a viscous fluid containing beads with 1 mm particles is not recommended with a gap of 1 mm.

You should reduce the number of decimals in several cases, e.g. instead of “357.28± 37.89” use “357± 38”. The result has to show the accuracy of the measurement method.

Reviewer 2 Report

As the authors have improved the manuscript to solve my previous concerns, this version could be acceptable.

Author Response

We thank the reviewer for accepting our revision.

Round 3

Reviewer 1 Report

The title is OK.
You have used the recommended equation in the newest version (Eq. 3.), but, please, correct the explanation of the characters, too.
Just a general advice for the future: Your diagrams should be more uniform in appearance!